# Training Postural Balance Control with Pelvic Force Field at the Boundary of Stability

**DOI:** 10.3390/bioengineering10121398

**Published:** 2023-12-06

**Authors:** Isirame Omofuma, Victor Santamaria, Xupeng Ai, Sunil Agrawal

**Affiliations:** 1Rehabilitation and Robotics (ROAR) Laboratory, Department of Mechanical Engineering, Columbia University, 500 West 120th Street, Rm 220, New York, NY 10025, USA; xa2117@columbia.edu; 2Department of Physical Therapy, School of Health Sciences and Practice, New York Medical College, New York, NY 10595, USA; vsantama@nymc.edu

**Keywords:** rehabilitation robotics, postural control, perturbation-based training, cable driven devices

## Abstract

This study characterizes the effects of a postural training program on balance and muscle control strategies in a virtual reality (VR) environment. The Robotic Upright Stand Trainer (RobUST), which applies perturbative forces on the trunk and assistive forces on the pelvis, was used to deliver perturbation-based balance training (PBT) in a sample of 10 healthy participants. The VR task consisted of catching, aiming, and throwing a ball at a target. All participants received trunk perturbations during the VR task with forces tailored to the participant’s maximum tolerance. A subgroup of these participants additionally received assistive forces at the pelvis during training. Postural kinematics were calculated before and after RobUST training, including (i) maximum perturbation force tolerated, (ii) center of pressure (COP) and pelvic excursions, (iii) postural muscle activations (EMG), and (iv) postural control strategies (the ankle and hip strategies). We observed an improvement in the maximum perturbation force and postural stability area in both groups and decreases in muscle activity. The behavior of the two groups differed for perturbations in the posterior direction where the unassisted group moved towards greater use of the hip strategy. In addition, the assisted group changed towards a lower margin of stability and higher pelvic excursion. We show that training with force assistance leads to a reactive balance strategy that permits pelvic excursion but that is efficient at restoring balance from displaced positions while training without assistance leads to reactive balance strategies that restrain pelvic excursion. Patient populations can benefit from a platform that encourages greater use of their range of motion.

## 1. Introduction

Postural control is foundational to carrying out everyday functional tasks, from standing to walking, running, reaching, and bending. It deals with manipulating the body by finely regulating its orientation and balance to perform desired functions [1,2,3]. Poor postural control can result in falls, devastating injuries, and even death [4,5]. Numerous people experience challenges with keeping a steady balance and could significantly benefit from postural control training. These include the elderly, individuals with neuromuscular conditions like stroke, those with spinal cord injury, patients with Parkinson’s disease, patients with dementia, and many others. Among the elderly, the Center for Disease Control and Prevention (CDC) has reported that one in four adults aged 65 years and older falls each year. Approximately 800,000 individuals in a year are hospitalized because of a fall injury and the total fall-related medical costs during the year 2015 totaled more than $50 billion [6]. There is thus a need to study training methods that are effective at improving postural control during upright standing in populations at-risk of falling.

Perturbation-based balance training is one method in rehabilitation used to improve balance control. It creates an environment to train balance through repeated exposure to destabilizing perturbations. Such training leads to improvements in reaction time, joint stability, and confidence during daily life interactions with the surroundings [7,8,9,10,11]. PBT has been implemented in several forms such as slip perturbations on treadmills [11,12] or through waist pulls [13,14,15]. In the work presented here, we perturbed subjects by applying an external force to their trunk to train their reactive balance control. Evidence in the literature has shown that training via repeated perturbations can lead to short-term adaptations [16]. Participants were perturbed repeatedly to give them the opportunity to generate and practice balance recovery strategies that could improve balance control [17,18]. This is also in alignment with the principle that practice is the most important factor in the improvement of motor skill [19].

Ideally, postural training methods promote learning of previously known and/or new postural control strategies to overcome challenging situations. A factor that influences motor learning is the challenge involved in the task [18]. Guadagnoll proposes that learning occurs when practice is done at the individual’s “challenge point” which is attained by taking into account the participant’s skill and the difficulty of the task [19]. They predict that beginners benefit more from training at lower task difficulty levels while experts have more potential to learn at a higher skill level [19,20]. In the context of postural control in standing, most healthy subjects can be classified as experts so we attempted to design a task that could challenge healthy subjects at a high skill level. This was achieved by modulating the difficulty of the task. Task difficulty was modulated by changing the magnitude of the perturbation, and adding a concurrent task, and using a blindfold [18].

Adding contextual interference to our training paradigm is another method that made the task more difficult and elevated it to the appropriate challenge point. This also increases the potential for learning and retention. The principle of contextual interference proposes that interference during practice is beneficial to skill learning, transfer, and retention [20,21,22]. Contextual interference is the interference experienced when practicing multiple skills or variations of a skill. Our VR postural training implements mass trial-and-error practice and contextual interference by applying perturbations in random directions and asking the subjects to complete a catch-and-throw task while receiving perturbations. This training setting has the participant practice each motor component of a serial motor task (catching and throwing while aiming at a target) from different postural configurations and at different levels of postural imbalance. These experimental conditions were set up to provide a challenge to the healthy participants of this study.

Repeated perturbations provide the opportunity for participants to practice balance recovery from an unstable position many times. When recovering from perturbations, subjects need to react quickly to avoid falling. We hypothesize that eliminating the requirement for quick reactions to avoid a fall, and providing a safe environment for the subjects to explore different recovery strategies will facilitate learning. The Robotic Upright Stand Trainer can create this safe environment by providing support to the subjects when perturbations pull them beyond their limits of stability. The hypothesis here is that the support provides the subject the opportunity to practice balance recovery strategies without the danger of falling. Without support beyond the limits of stability, either subjects lose balance when in unstable positions or they develop strategies to avoid reaching those unstable positions. Training with support should provide the context for subjects to become familiar with restoring balance from these unstable positions which could translate to increased resistance to perturbations.

Thus, in this paper, we ask the question: Does training with assistive forces at the stability limits change the outcomes of postural balance training? Does it improve balance control?

Training was conducted in RobUST, a cable-driven exoskeleton developed in the ROAR lab at Columbia University, Figure 1 [23,24]. It consists of two end-effector braces, each connected to four cables distributed around the brace circumference. Each cable is connected to a motor and thus each brace can be actuated in the horizontal plane. The system offers unique advantages in the training of postural standing by permitting weight bearing and only intervening when assistance is needed. It thereby encourages the improvement and not replacement of body function. In the device, a subject is able to train in close to real-life situations with some security provided by the device.

## 2. Materials and Methods

As depicted in Figure 2, the present study followed a quasi-experimental cross-sectional pretest-posttest design with two groups. It contained a PBT intervention where one group had assistive support provided by RobUST’s pelvis belt during the training sessions and the other group did not have any support. Force assistance was implemented as a force field around the subject. The group with support is referred to in the text as the Force Field group (FF) and the group without support as the No-Force Field group (NF). Balance characteristics were assessed before and after the intervention. The contents of the pretest, posttest, and intervention are explained later in the paper.

All subjects consented to participate in the experiment. The experiment was reviewed and approved by the Columbia University Institutional Review Board under protocol number IRB-AAAR6780. A total of 10 healthy adults were recruited (5 male, 5 female; 7 right-handed, 3 left-handed; between 21 and 40 years old; mean weight = 62.4 ± 8.34 kg, and mean height = 1626 ± 65 mm). Participants were randomly assigned to one of two groups: a group with assistive force field at the pelvis (FF) (*n*= 5), and a second group without force-field assistance (NF) (*n* = 5). The participants remained oblivious to the type of intervention received.

Perturbation-based training was done in RobUST and subjects were perturbed at their trunk in the forward, backward, right, and left directions. All perturbations had a short trapezoidal profile with 150 ms ramp up, 300 ms at the set force, and 150 ms decay time.

Subjects in the FF group were given force field support at their pelvis only during the intervention. The force field was an assist-as-needed setting in which RobUST applied no supportive force when the subject was within their limits of stability. A subject’s limits of stability is a polygonal area within which the subject can be displaced without losing balance. RobUST applied a force directed towards the neutral starting position through its pelvic belt once the subject’s pelvis moved outside the stability limits. We describe how this area is determined in the next section.

The experiment comprised three parts: (i) Pretest, (ii) Virtual Reality training, and (iii) Posttest.

Pretest: The subjects’ pre-intervention balance was characterized in this session. To do this, subjects were blindfolded [25] and perturbed without pelvic assistance 15 times in each direction starting at a force of 40% of the body weight (BW). The modified 4-2-1 algorithm [26] was used to progress the perturbation force. Subjects were perturbed 15 times, regardless of whether the 4-2-1 algorithm reached the threshold before 15 tries or not. The primary outcomes measured were the force threshold and the stability area. The steps of the algorithm are as follows:(a)Start perturbations at 40% BW and increase perturbation force by 4% BW after all successful trials until the subject loses balance. A successful trial is a trial in which the subject did not change their base of support, i.e., no stepping, foot sliding, or reaching during the trial.(b)After the first time the subject loses balance, decrease force by 2% and keep decreasing until the next successful trial.(c)Once the subject is able to maintain balance again, increase the perturbation force by 1%.(d)Increase or decrease the perturbation force by 1% in all subsequent successful or failed perturbation trials respectively.(e)The average of the three highest force perturbations in each direction is taken as the force threshold for that direction.The “stability area” of the pelvic center was measured using only successful trials. This area was calculated by collecting all the points through which the subject moved after being displaced from their starting position by a perturbation. An external boundary that encloses these points was then drawn and this boundary was established as the subject’s stability limits. The area within the boundary was also calculated as the stability area. In summary, from the pretest, we determined the force threshold in each direction and the stability region. These two balance measures were used in subsequent sessions.Virtual Reality Training: We used a VR game developed in the ROAR Lab and run in Unity on an HTC Vive (HTC Corporation, Taoyuan City, Taiwan) VR headset for the experiment (Figure 1 and Figure 3). The subject had to catch a ball projected at their chest, and then aim and throw the ball at a moving target to score points. The ball was programmed to reach the chest of the subject in one second and so RobUST was set to randomly deliver trunk perturbations between 0 and 0.8 s after the ball was tossed towards them so that the subject’s catching was perturbed. The force of the perturbations delivered in each direction corresponded to the force threshold established during the pretest phase. The FF group received 15% BW restoring force from the pelvic belt if they strayed outside their stability limits when catching and throwing while the NF group did not receive this assistance. The restoring force assisted the subject to return to the neutral standing position. Before each perturbation, the subject was given enough time to reset to a neutral position, and the stability limits, found during the pretest, was centered on this neutral standing position. The training session was divided into two sessions and the catch-throw-perturbation sequence was repeated 50 times in each session.Postest: The posttest was identical to the pretest in which the force threshold and stability area were measured. The subjects took a 5-min rest before the posttest.

### 2.1. Data Collection

Kinematic data of the human body segments were collected using a Vicon motion capture system (Vicon, Denver, CO, USA) with nine cameras (Vicon Vero 2.2) that picked up retroreflective markers worn by the subjects. The frame rate for data capture was 100 Hz and kinematics were processed offline using Vicon Nexus software 2.10.0 and MATLAB R2019b (MATLAB, Natick, MA, USA). A biomechanical model was constructed based on optical markers placed on the subject. One marker was placed on the left and right shoulders, three on the trunk belt, and three on the pelvic belt. Three markers were placed on each foot—on the nail of the big toe, the fifth metatarsophalangeal joint, and the heel. Participants had a starting position with each foot on a six-axis force plate (Bertec, Columbus, OH, USA). The force plate provided the ground reaction force and the location of the combined and individual foot center of pressure at a 1000 Hz frame rate. Data from the force plates were synchronized with motion capture data in the Vicon Nexus 2.10 software. The software also provided the combined COP obtained from each one of the force plates. Electromyographic data were also collected bilaterally using 14 channels of a Delsys Trigno Wireless System (Delsys Incorporated, Natick, MA, USA) from the following muscles at 1000 Hz: tibialis anterior (TA), lateral gastrocnemius (LG), rectus femoris (RF), bicep femoris (BF), gluteus medius (GM), rectus abdominis (AB), and erector spinae (ES). In naming muscles in this paper, we affix the side of the EMG to the muscle on which the EMG sensor was placed, e.g., we refer to the left TA as L-TA and the right TA as R-TA.

### 2.2. Variables Measured

There were three sets of data at the end of this experiment: (i) gross performance measures (ii) kinematic measures; (iii) muscle activity measures.

#### 2.2.1. Gross Performance Measures:

We collected the force thresholds and the stability area during the balance tests. These were described at the beginning of Section 2. In addition, during the virtual reality sessions, we collected the trial success rate, the proportion of trials in which the subject did not lose balance, and the number of successful catches.

#### 2.2.2. Kinematic Measures

The margin of stability (MOS), the center of mass (COM) excursion (pelvic excursion), the center of pressure excursion, and COM velocity were calculated as measures of the subject’s movement during each trial. A trial was defined as a two-second period starting when a perturbation was initiated. A marker on the subject’s sacrum was used as an estimate of the COM position so COM and pelvis position are used interchangeably in this text.

##### Margin of Stability (MOS)

MOS was defined as the minimum distance of the extrapolated center of mass (COM) from the perimeter of the base of support (BOS) during each trial, Figure 4 [27]. Optical markers on the big toe, fifth metatarsal, and heel of both feet were used to create the BOS outline.

##### COM and COP Measures

The COP and COM total excursion, COP_TE and COM_TE respectively, are the length of the path moved through by each during a perturbation trial. The maximum displacement from the starting position was recorded as COP_Max and COM_Max, their maximum velocities as COP_Vel_Max and COM_Vel_Max, and their mean velocities as COP_Vel_Mean and COM_Vel_Mean.

##### Balance Strategies

Balance strategy refers to the use of either the ankle or hip strategy in response to a perturbation [28]. We divided the body into an upper body trunk segment and a lower body segment (thighs, shank, and feet) and examined correlations between the upper body angle in the sagittal plane and the lower body angle. The ankle strategy was identified by positive correlations between the upper body and lower body while the hip strategy was identified by negative correlations between these segments, Figure 5 [29]. This gives us the following classification for upper-body and lower-body rotations (Table 1).

Correlations were performed using a 250 ms centered moving window at each time point over the 2 s perturbation trial. The critical value/threshold for significance of R-values for 25 data points (position data was recorded at 100 Hz) at 0.05 significance level is 0.3807. Subjects were said to be using the ankle strategy for *R*-values above 0.3807, and the hip strategy for *R*-values below −0.3807, Figure 6. *R*-values in between −0.3807 and 0.3807 were left undefined. Once strategies were identified, the percentage of the 2 s perturbation trial spent using the ankle strategy and the percentage time in hip strategy were calculated and recorded as Ankstrat and Hipstrat respectively. These were used as the assessment variables.

##### Muscle Activity Data

EMG signals were detrended, band-pass filtered between 20 and 300 Hz, full-wave rectified, and low-pass filtered at 20 Hz. The processed signal was used to calculate integrated EMG (iEMG) for each perturbation trial and the unprocessed signal was used to calculate EMG activity duration (EMG_DUR).

iEMG was calculated as the summation of muscle activity over the perturbation trial and it was calculated and normalized using the formula:(1)$$iEMG=\frac{\sum_{onset}^{onset+t}EMG-\sum_{t}EMG_{baseline}}{t\times max(EMG_{global})}$$
where $\sum_{onset}^{onset+t}EMG$ is the summation of the EMG signal from perturbation onset over a *t* = 2 s time window; and $\sum_{t}EMG_{baseline}$ is the summation of a baseline EMG signal while the subject is in quiet standing over 2 s. $max(EMG_{global})$ is the maximum observed EMG value per subject.

EMG activation was calculated from detrended EMG data passed through the Teager-Kaiser energy operator (TKEO) [30], rectified and low-pass filtered at 50 Hz. The activation threshold was set to the baseline TKEO mean plus three standard deviations. A muscle was considered to be active when its TKEO signal rose above this activation threshold for more than 50 ms and inactive when the muscle signal fell below the threshold for more than 50 ms after being previously active. Activity duration was the sum of all intervals between muscle activations and deactivations.

### 2.3. Statistical Analysis

Statistical tests to detect the effect of our VR training program with and without assistive pelvic forces, and the difference between training in the NF group and the FF group were set up using the statistical software SPSS, (IBM, version 26, 2019). Only successful trials, defined as those trials in which the subject did not lose balance, were used in this analysis. Thus, tests to detect the main effect of training, of training type, and of an interaction effect between the two were set up.

A two-factor mixed Analysis of Variance (ANOVA) with one within (pretest vs posttest) and one between factor (NF vs FF groups) was used. Data was tested for violation of sphericity and homoscedasticity with Mauchly’s test and Levene’s test, respectively. Post-hoc analysis were performed after applying Bonferroni’s inequality procedure to correct for multiple comparisons. Interaction effects were prioritized to distinguish any relevant postural effects in the group that received the assistive force field. However, if interaction effects were absent, the main effects were interpreted.

To examine the frequency of stepping (or not stepping) during the VR training session, we applied a non-parametric Mann-Whitney U test to this data as it was not normally distributed. Since the use of force field was hypothesized to have only a positive effect, the statistical test was one-tailed. The statistical variables examined were the median, minimum, and maximum stepping frequencies of the NF and FF groups.

The range of perturbation forces differed between pretest and posttest as seen in the distribution of successful trials, Figure 7 and the change in perturbation force threshold values Figure 8. In order to compare metrics within the same range, perturbation forces above 50% BW and below the lower of the pretest force the posttest force thresholds were selected. Figure 7 shows the number of valid trials for each subject used to calculate the results.

## 3. Results

### 3.1. Gross Performance Measures

#### 3.1.1. Force Thresholds and Stability Area

The force thresholds, across all four directions significantly improved in the FF and NF groups: posterior (F(1,8)=6.51;p=0.03), anterior (F(1,8)=6.37;p=0.04), left (F(1,8)=19.82;p=0.02), and right (F(1,8)=18.14;p=0.003), Figure 8. The same was true for postural stability area in standing (F(1,8)=13.63;p=0.006) (Figure 9).

#### 3.1.2. Virtual Reality Training

Differences in motor performance between the NF and FF groups were observed during the training session. The number of successful catches in the FF group (median = 50, IQR = 3) was significantly higher than that in the NF group (median = 48, IQR = 14) (U=6.5;p=0.03). The number of times the subjects lost balance tended to be higher in the NF group although it was not significantly different (*U* = 3.0 from the FF group. NF group median = 8 and FF group median = 1, p=0.1).

### 3.2. Kinematic and EMG Measures

Posterior Perturbations: There was an interaction effect during posterior perturbations for the variables shown in Table 2 and Figure 10. For COM_Max, there was a significant decrease in the NF group, an increase in the FF group, and a resulting significant difference in the posttest of the NF and FF groups. For COM_TE, there was a significant increase for the FF group and a significant difference between the posttest averages of the two groups. For COM_Vel_Max, there was a significant speed increase in the backward direction in the FF group which resulted in a significant difference in the posttest of the two groups.

Simple main effects analysis showed that training did not have a statistical effect on MOS (p=0.384) but did have significant group training effects that resulted in increases in R-LG DUR (p=0.038), and decreases in L-AB iEMG (p=0.030), L-TA iEMG (p=0.024), L-BF iEMG (p=0.043), R-TA iEMG (p=0.007).

Statistically significant changes were observed in the forward, right, and left directions but these changes were primarily reductions in the muscle activity that occurred in both groups and do not say much about the effect of the different training methods. We therefore will focus on the results of the posterior perturbations.

## 4. Discussion

In this study, we examined the effects of a robot-mediated PBT with RobUST on two groups of healthy adults: one group with postural assistance via force fields, and the other group without assistance. Overall we found that the two training methods led to significant improvements as indicated by the ability to resist higher trunk perturbations. The primary evidence of improvement in both groups was increased force thresholds (the maximum force a subject is able to resist before losing balance) between the pretest and the posttest for all perturbation directions, Figure 8. In addition, increased stability area was observed, Figure 9.

Our data on MOS, COM excursion (pelvic excursion), and balance strategy also revealed that our training modalities can be delivered to promote the learning of distinct postural control strategies. The assisted group learned to control posture by permitting more pelvic excursion; whereas, the unassisted group permitted less excursion. Evidence of differences in adaptation between NF and FF groups was observed primarily in the most unstable postural direction (i.e., posterior direction) where the MOS of FF group subjects after training was lower, their COM excursion was higher and NF group subjects used the ankle strategy less often.

### 4.1. Balance Strategies

Our assessment of balance strategies plays a key role in our understanding of the postural adjustments made in the presence and absence of assistive force fields. We were able to identify patterns around two pivotal joints in postural standing, the ankle and the hip. Perturbations were enacted as a pull on the trunk lasting 600 ms. During this time, the subject’s COM was forcibly moved in the direction of the pull leaving them close to the boundaries of their BOS at the end of the perturbation. At this point, the COP-COM distance was minimized and thus, the ability of subjects to generate any restoring torque using the ankle strategy was minimal because the COP-COM distance is directly proportional to the restoring torque [31,32]. In the first few hundred milliseconds of the perturbation, when the COP-COM distance is maximal, we observed that subjects tried to use an ankle strategy but quickly changed to the hip strategy once the perturbation started. Once they regained balance, moving their COM within their BOS, they used the hip strategy less. This was reflected by a reduction in the average ratio of hip strategy percentage to ankle strategy percentage between the beginning and end of the perturbation response. In the first 500 ms of reaction, across all conditions and subjects, the average ratio of hip strategy percentage to ankle strategy percentage was 2.4:1 and in the following 500 ms, it was 1.4:1.

While analyzing the movement of subjects in response to perturbations, some common reactions were identified. In responding to posterior perturbations, subjects flexed the hip to bring the trunk forward as the perturbations forced trunk extension. We observed that subjects varied in the timing and direction of movement of their lower body. Some subjects started flexion at the hip very early for lower forces while others only attempted hip flexion when exposed to higher forces. This difference could be because some had the required flexibility to flex the hips during trunk extension and others did not. Anterior perturbations on the other hand forced trunk flexion. Subjects initially complied with the perturbation by following with hip flexion in the first few hundred milliseconds but eventually, they resisted by initiating hip extension. An additional mechanism used by subjects to respond to posterior perturbations was the fast dorsiflexion of the feet. We postulate that this technique was employed because dorsiflexion, viewed from the right of the subject, is the counterclockwise rotation of the feet which applies a clockwise moment to the body. The clockwise moment has the effect of moving the body forward, moving the COM within the BOS, and towards stability. Similarly, subjects responded to trunk flexion caused by anterior perturbations by plantarflexing and rising onto their toes. These two mechanisms—hip flexion and foot dorsiflexion for posterior perturbations, and the corresponding reactions to anterior perturbations—are part of the hip strategy, where the body internally generates moments that move it towards stability.

### 4.2. Effects of Perturbation Training

Subjects in our study completed a single training session and improved their tolerance of perturbative forces in all perturbation directions. Sturnieks et al. [14], in a waist-pull experiment, showed that the value of the force threshold is a strong indicator of the robustness of postural balance control in older adults. Generating threshold increases in healthy subjects points to balance control improvements in these subjects and suggests the potential for RobUST to generate these improvements in patients with balance disorders.

The subjects increased their stability area, that is, the region within which they maintained balance while being perturbed. Previous studies have highlighted a positive relationship between steady postural sway in standing and an increased risk of falls [33,34,35,36,37,38]. But in our study, the stability area was a measure of the extent to which the pelvis could be pulled away from the subjects’ neutral positioning without losing balance. It differs from postural sway during steady standing. In this study, COM movement was induced as a result of trunk perturbations and our finding that this variable increased in the posttest indicates that subjects can withstand larger perturbation forces without losing balance after the completion of a session of PBT with RobUST. In [39], where older adults were subjected to an 8-week training program that included combinations of strength, flexibility, static and dynamic balance exercises, an increase in mediolateral sway resulted in an improvement in dynamic functional postural task performance, such as the Timed Up & Go Test. The participants of that study learned to control their balance over a greater range of motion, and this translated to an expanded standing workspace during dynamic balance conditions. In the same way, we assume that the observed increase in stability area in our experiment could translate to an improvement in the ability of the subjects to maintain balance. However, further studies including balance tests are needed to corroborate this hypothesis.

Reductions in muscle activity, identified as reductions in muscle iEMG and max EMG amplitude, were observed along with this increased level of performance. There is evidence in the literature that repeated exposure to perturbations results in more efficient postural strategies in which individuals change from a more vigorous to a less vigorous response, and fewer or different muscle fibers are recruited [8]. Specific observations in our study support this; for example, in the left direction, R-TA, R-LG activation duration reduced (Appendix A, Table A1) as well as iEMG of L-TA, L-LG, L-BF and R-LG and the max amplitude of L-TA, R-LG, L-BF, and R-BF. Similar reductions were observed in the anterior and right directions. Muscle activity reductions and increased force threshold observed can thus be interpreted as improvements in balance control accompanied by a decrease in energy expenditure.

We did not observe differences between the thresholds for the two groups which may be due to a ceiling effect, i.e., the threshold and stability area measures were at maximum levels after PBT of healthy subjects and, so more subtle differences in training methods could not be discerned from these variables.

### 4.3. Effect of Perturbation Training with RobUST Force Field

In developing our experiment’s protocol we hypothesized that enabling subjects to spend more time recovering balance from the limits of stability would improve their stability and we tested this by training a group with RobUST force assistance and another group without force assistance. The results showed that the FF group developed a strategy in the posttest session characterized by reduced MOS and increased COM excursion for posterior perturbations while the NF group had more MOS, less COM excursion, and less use of the ankle strategy. The FF group thus, developed a strategy in which they moved closer to the stability limits while still being able to maintain balance, and the NF group adapted towards less excursion. The two groups however, did this while still maintaining the same perturbation success rate and even increasing their perturbation force thresholds in the posttest.

The difference in postural control strategies adopted by the two groups to overcome the perturbation while achieving the VR task goal for posterior perturbations is of interest. We hypothesized that training at the stability limits, augmented by assistance that promoted time spent at the limits would result in improved balance performance. Our experimental results show that the FF group subjects permitted more excursion and they learned to restore balance from positions closer to the stability limits. Although this did not translate to a significant increase in posterior perturbation force threshold, it does mean that FF group subjects increased the range of motion in the posterior direction and that they learned to recover stability from a more displaced position. We may be training this group to learn recovery postural strategies before the COM crosses the BOS boundaries and a stepping strategy is required.

Increasing the range from which a subject is able to recover balance increases the number of situations in which the subject is able to maintain balance in the presence of unexpected disruptions, hence improving postural stability. Increasing this range of motion is one of the aims of rehabilitation [40]. Training in this paradigm facilitated both increased perturbation resistance and increased excursion for the FF group. If these results can be transferred to patient populations, our training paradigm could be used to better achieve the aims of balance training for patients.

## 5. Conclusions

In this experiment, we tested the sensorimotor adaptation of the postural control system. We found that subjects improved balance after training postural control in a VR game in the two experimental conditions tested, training with RobUST force assistance and training without. Even though both groups improved balance, differences were observed in MOS, COM excursion and ankle strategy percentage parameters suggesting that the two groups adapted differently. The FF group applied a balancing strategy with less MOS and increased COM excursion than the NF group and thus is likely they were more adroit at recovering balance from displaced positions. The increases in force threshold and stability area, and reductions in muscle activity in all subjects show that training with RobUST assistance generated improvements in balance performance and it is thus a viable platform for balance training. Training without assistance is a more difficult training scenario and healthy subjects responded to this by limiting their COM excursion. From these results we can conclude that training with RobUST assistance promoted a balance recovery strategy that permitted recovery from positions of greater displacement for perturbations in the posterior direction. This skill could be useful when responding to unexpected perturbations.

Finally, although we showed increases in force thresholds in both groups, the improvements were not directly related to increases in muscle activity. Decreases in muscle activity suggest the improved performance may be a result of improved muscle coordination, i.e., less muscle agonist-antagonist co-activation, and more efficient muscle synergies. In addition, we suspect changes to core, back and glute muscles, and possibly muscle activity increases contributed to increased resistance to trunk perturbations. Core and back muscles were difficult to record because of the presence of adipose tissue on the subjects.

RobUST assistance was effective at improving pelvic excursion because it was a safe and stable training condition in which subjects could explore pelvic movements and thus generate balance recovery strategies from displaced positions. We propose that this training paradigm can be used to increase hip range of motion in subjects where that is limited e.g., SCI and cerebral palsy.

## 6. Limitations

The primary limitation of this work was that the test group consisted of healthy adults and not patients challenged by a balance disorder. This leaves room for doubt about whether gains seen in healthy patients will also be seen in patients with disorders.

The processing of experimental results was beset by low amounts of data. There was data loss because of the malfunction of recording devices. The protocol for establishing thresholds ensured that subjects were not tested within a narrow range of BW percentages. Data loss and the wide range of perturbation forces led to uneven amounts of low and high perturbation force data in some subjects and we accounted for this by selecting data only within 50 and 75% BW. This range was chosen to have a substantive amount of trials from each subject but it using such a wide range %BW introduces inaccuracies in our summary variables as each subject could be reacting to a different magnitude of forces on average.

We concluded that training with RobUST forcefield improved balanced recovery from displaced positions. To further confirm these results, an alternative testing design would be to position subjects at a distance from their neutral and ask them to move back to neutral. We would then have definitive proof that subjects were more capable of restoring balance from larger displacements.

We have not tested the long-term training effects due to our robotic-VR training in this current study but we hope to explore this aspect in future studies.

## Figures and Tables

**Figure 1 bioengineering-10-01398-f001:**
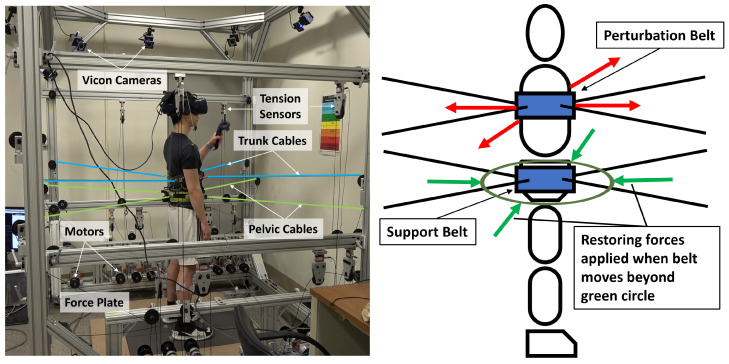
Subject standing in RobUST. (**Left**) Subject in RobUST with cables attached to trunk and pelvic belts, wearing a virtual reality headset, and holding a VR controller. The subject stands on two force plates that measure ground reaction forces. A Vicon motion capture system was used to record the subject’s movement. (**Right**) Body diagram in our experimental setup displaying belts (blue rectangles), perturbation trunk forces (red arrows) assistive force field (green circle) and assistive pelvic forces (green arrows).

**Figure 2 bioengineering-10-01398-f002:**
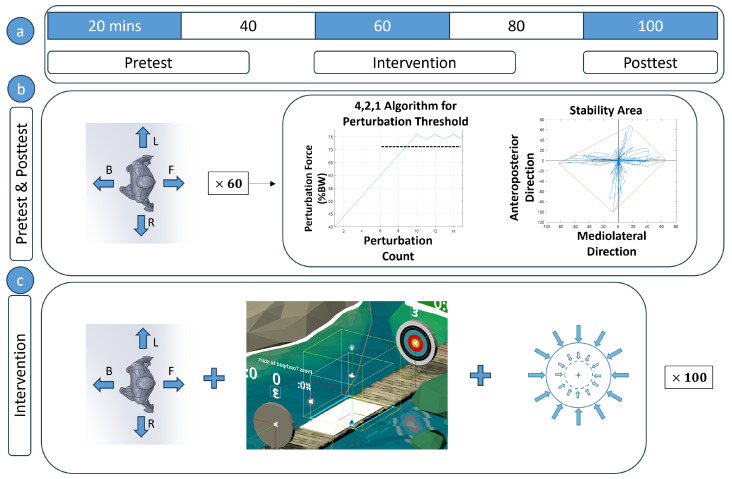
(**a**) Experiment procedure and timeline (**b**) The pretest and posttest were conducted in a similar fashion. Subjects were perturbed 60 times, (15 in the forward (F), backward (B), left (L), and right (R) directions) and the perturbation magnitude was progressed using the 4,2,1 algorithm. The perturbation force threshold and stability area were found during these tests. (**c**) In the intervention, subjects were perturbed while playing a virtual reality catch-and-throw game. A group of subjects had given force assistance and some did not. Force assistance was activated once the subject moved outside a boundary and continued till the subject was a negligible distance from the subject’s starting position.

**Figure 3 bioengineering-10-01398-f003:**
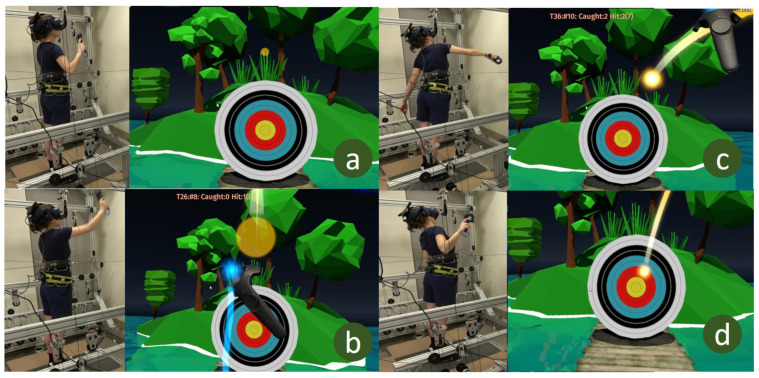
Subject and view from VR game. (**a**) Subject prepares to receive the ball, (**b**) subject catches the ball, (**c**) subject throws the ball, (**d**) subject tries to recover balance after throwing the ball at the target.

**Figure 4 bioengineering-10-01398-f004:**
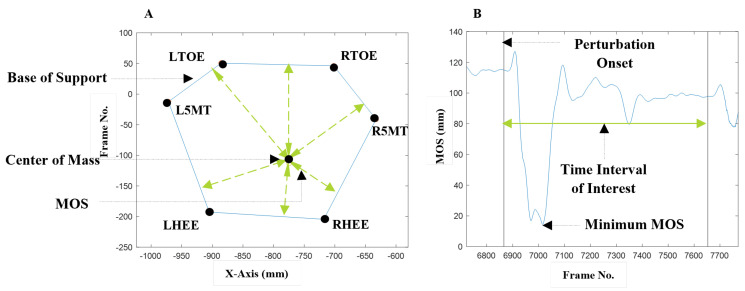
(**A**): Center of mass (COM) shown within the base of support (BOS). Markers on the feet, RTOE—Right Toe, R5MT—Right Fifth Metatarsal, RH—Right Heel, LT—Left Toe, LM—Left Fifth Metatarsal, LH—Left Heel, form the BOS. The margin of stability (MOS) at each time point is the minimum distance of the COM to the BOS. (**B**): The trial MOS is the minimum value of the margin of stability across all time points in the trial.

**Figure 5 bioengineering-10-01398-f005:**
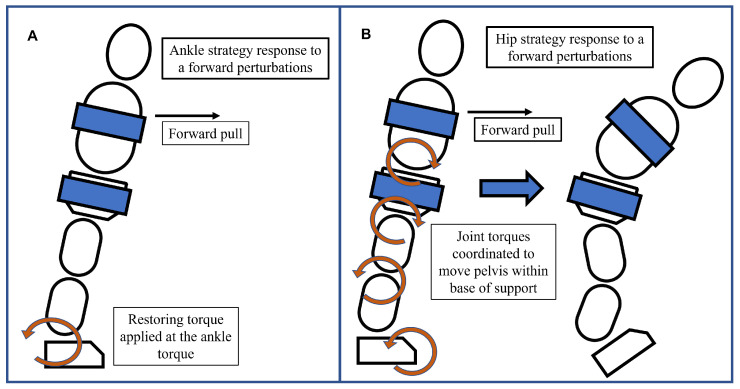
(**A**) Ankle strategy response to a forward pull. In this response, the upper and lower body rotate in the same direction about the ankles like an inverted pendulum while other joint positions are held rigid. Restoring torques are applied at the feet. (**B**) Hip strategy response to a forward pull. Restoring torques are generated at the joints, which manipulate body parts and move the center of mass back within the base of support. Hip flexion and foot dorsiflexion create dynamic moments that move the body and attempt to restore balance.

**Figure 6 bioengineering-10-01398-f006:**
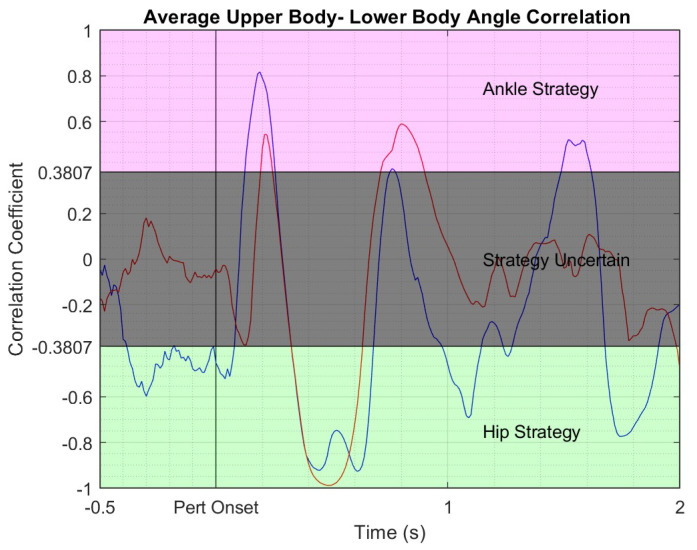
Correlation of upper and lower body sagittal orientation averaged across perturbation trials for a representative subject in the NF group. Blue line—pretest correlation; red line—posttest correlation. Pink shaded area—region of ankle strategy where upper and lower body rotate in the same direction and move like an inverted pendulum around the ankles; grey shaded area—area of transition between ankle and hip strategies; green shaded area—hip strategy region where the upper and lower body rotate in opposite directions around the hips.

**Figure 7 bioengineering-10-01398-f007:**
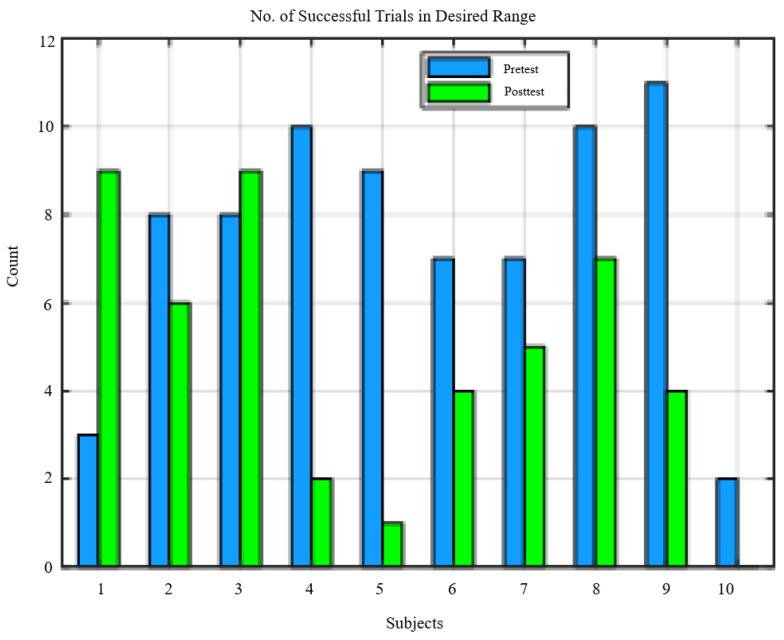
Distribution of successful trials per subject.

**Figure 8 bioengineering-10-01398-f008:**
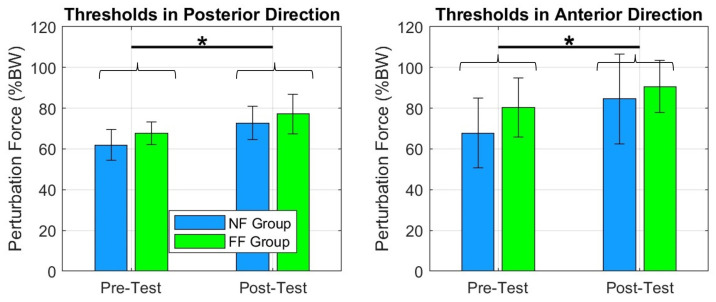
The plots show an increase in force thresholds between pre- and posttests for FF and NF groups across the four directions. We can notice that both groups improved their tolerance to receive greater perturbation intensities. * = p≤0.05.

**Figure 9 bioengineering-10-01398-f009:**
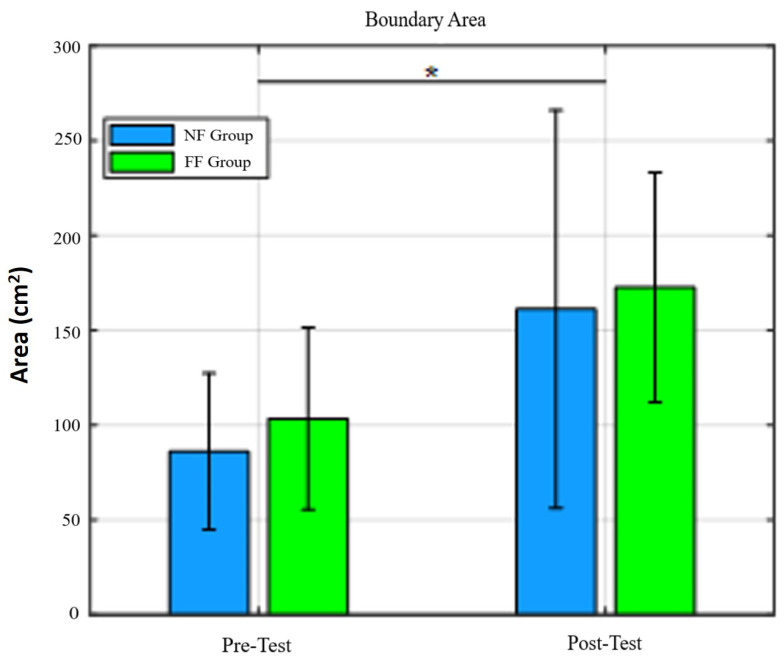
Stability area for pretest and posttest values in NF and FF groups. We can observe that RobUST, in combination with VR, was enough to increase the area of postural stability in standing. * = p≤0.05.

**Figure 10 bioengineering-10-01398-f010:**
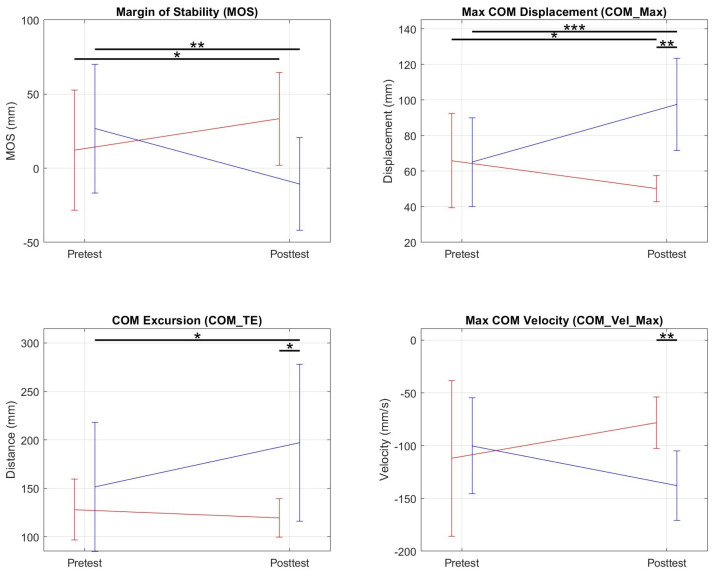
Graphs of pretest and posttest averages of NF (red) and FF (blue) groups for metrics in which a significant interaction effect was present. For the margin of stability, we found a significant increase between the pretest and posttest for the NF group and a decrease for the FF group. * = p≤ 0.05; ** = p≤0.01; *** = ≤ 0.001.

**Table 1 bioengineering-10-01398-t001:** Results of correlation for different combinations of upper body and lower body rotations.

	Anterior Perturbation	Posterior Perturbation
Ankle Strategy	Trunk forward rotation—Lower body forward rotation	Trunk backward rotation—Lower body backward rotation
Hip Strategy	Trunk forward rotation—Lower body backward rotation	Trunk backward rotation—Lower body forward rotation

**Table 2 bioengineering-10-01398-t002:** Statistics for variables with significant interaction effects and the corresponding post hoc tests for trials in the posterior direction.

Variable	DOF	*F* Statistic	*p*	FF Pre vs. Post	NF Pre vs. Post
MOS	6	8.815	0.025	↓p=0.028	−p=0.300
Ankstrat	7	8.813	0.021	−p=0.133	↓p=0.001
COM_Max	8	29.67	0.001	↑p<0.001	↓p=0.036
COM_TE	8	7.856	0.023	↑p=0.010	−p=0.553
COM_Vel_Mean	8	7.111	0.029	↑p=0.014	−p=0.550
COM_Vel_Max *	8	9.327	0.016	−p=0.052	−p=0.076

↑ signifies increases in the value of the metric averaged across all subjects. ↓ signifies a decrease in the value of the metric. − signifies no change in the value of the metric. * No significant change was observed between pretest and posttest in both FF and NF groups but the posttest averages of the groups differed in the post hoc analysis (*p* = 0.009). NB: Significant results in the anterior, right and left directions can be found in Appendix A.

## Data Availability

Derived data supporting the findings of this study are available from the corresponding author, I.O., on request.

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
