# Peer review of "Training Postural Balance Control with Pelvic Force Field at the Boundary of Stability"

_bioengineering, 2023, doi:10.3390/bioengineering10121398_

Round 1

Reviewer 1 Report

Comments and Suggestions for Authors

The study title is focused on the Training Postural Balance Control with Pelvic Force Field at the Boundary of Instability. This is preliminary study because the sample size is very small and based on the healthy population. My proposition is to consider future study with the larger and different age population. RobUST assistance is effective for balance training, promoting a specific balance recovery strategy, and improving muscle coordination. Please add how it is useful. Consider breaking down some of the longer sentences into smaller, more digestible ones for clarity.

Comments on the Quality of English Language

The study title is focused on the Training Postural Balance Control with Pelvic Force Field at the Boundary of Instability. This is preliminary study because the sample size is very small and based on the healthy population. My proposition is to consider future study with the larger and different age population. RobUST assistance is effective for balance training, promoting a specific balance recovery strategy, and improving muscle coordination. Please add how it is useful. Consider breaking down some of the longer sentences into smaller, more digestible ones for clarity.

Author Response

Thank you for taking the time to review the manuscript and for pointing out the need for clarity in writing and in stating our end goal. Please see how we have addressed your comments below:

Reviewer Comment: The study title is focused on the Training Postural Balance Control with Pelvic Force Field at the Boundary of Instability. This is preliminary study because the sample size is very small and based on the healthy population. My proposition is to consider future study with the larger and different age population.

Author's Response: Thank you for your recommendation. We are in the process of considering training with the spinal cord injury population.

Reviewer Comment: RobUST assistance is effective for balance training, promoting a specific balance recovery strategy, and improving muscle coordination. Please add how it is useful.

Author's Response: RobUST assistance was effective at improving pelvic excursion because it was a safe and stable training condition in which subjects could explore pelvic movements and thus generate balance recovery strategies that involved this movement. We propose that RobUST can be used to improve balance training in populations like spinal cord injury where the aim of training is to increase the range of motion of the subjects. Following from your inquiry, the conclusion of the paper has been updated to reflect this idea.

Reviewer's Comment: Consider breaking down some of the longer sentences into smaller, more digestible ones for clarity.

Author's Response: Thank you for this comment. We appreciate the need to communicate clearly and have read through the document. We simplified some of the sentences and corrected some typographical errors.

Reviewer 2 Report

Comments and Suggestions for Authors

There are some format problems in the text mainly when presenting measures (value unit - written without space) which are signed in the file attached.

In line 240 there is a reference missing ( but could be presented without it if is not relevant to the discussion).

In lines 263 and 264 there are some lines in bolt that should be reformatted.

The titles in Table 2 and Tables A1 aA2 could reformulated for a clearest reading. And also some figures in order to have straight columns.

Figures 8, 9 and 10 use stars annotations that meaning are not clear in the presentation and discussion of the results. The vertical labels of Figures 8 and 9 should be edited to be in the format - variable (unit). The same problem exists with the horizontal label in Figure 4.

The legend of Figures 6 and 10 is too long. The prepositions made there should be shown and articulated within the text.

Comments on the Quality of English Language

I have no special comments about the English.

Author Response

Thank you for taking the time to review the manuscript and for pointing out our formatting errors. This goes a long way in ensuring the quality of our paper. Please see how we have addressed your comments below:

Reviewer’s Comment: There are some format problems in the text mainly when presenting measures (value unit - written without space) which are signed in the file attached.

Authors’ Response: Thank you for your detailed review of our formatting and for including the file identifying the errors. We have made the corrections identified and further read through the text to catch any other errors.

Reviewer’s Comment: In line 240 there is a reference missing (but could be presented without it if is not relevant to the discussion).

Authors’ Response: Thank you for catching this omission. The reference has been added.

Reviewer’s Comment: In lines 263 and 264 there are some lines in bolt that should be reformatted.

Authors’ Response: The lines in bold have been made normal

Reviewer’s Comment: The titles in Table 2 and Tables A1 aA2 could reformulated for a clearest reading. And also some figures in order to have straight columns.

Authors’ Response: Thanks for your suggestions for improving the table. We have reformatted Table 2 by adding columns for the degree of freedom and p value of the F statistic so that it is more readable.

Reviewer’s Comment: Figures 8, 9 and 10 use stars annotations that meaning are not clear in the presentation and discussion of the results. The vertical labels of Figures 8 and 9 should be edited to be in the format - variable (unit). The same problem exists with the horizontal label in Figure 4.

Authors’ Response: Thank you for thoroughly reviewing the paper. The absence of a description for the star annotation was an omission. They have been included. The formatting of the axis labels has also been updated.  

Reviewer’s Comment: The legend of Figures 6 and 10 is too long. The prepositions made there should be shown and articulated within the text.

Authors’ Response: Thank you for your comment. The caption of Fig. 6 fully describes the color coding used and we think that that description should be in the caption of the figure rather than in the paper’s text. The detail in Fig. 10 has been moved to section 3.2 of the paper.

Reviewer 3 Report

Comments and Suggestions for Authors

The manuscript “Training Postural Balance Control with Pelvic Force Field at the Boundary of Instability” by Isirame Omofuma et al reports an improved strategy for postural control training. Perturbation-based balance training is used in rehabilitation to improve balance control. The authors have developed a device which helps to restore the pelvis position after perturbation if its deviation is too large. They hypothesized that the assistance at the limits of stability would improve training outcomes because subjects may safely explore different recovery strategies. The authors also used a concurrent VR task to make the perturbations more challenging (to bring the subject posture to the limit of stability).

The manuscript is solid and well-written. However, I have questions to be answered before publication.

Major points:

“Correlations were performed using a 250ms centered moving window at each time point over the 3s perturbation trial”. Please elaborate how the edge effects were managed when computing correlation near 0s and 3s time points?

Figure 6: please specify the group of the representative subject. In the figure, the red plot resembles the blue one shifted earlier in time by approximately 200 ms. Does this mean that the subject’s reaction time was faster during the posttest than during the pretest?

Section 3.1.2: “The number of successful catches in the FF group (median = 50) was significantly higher than that in the NF group (median = 48)”. Additional comments are needed because the numbers are actually very close. I would say that the numbers of successful catches are not significantly different between groups.

If I understood correctly, Table 2 is for Posterior Perturbations only. What about data on other perturbation directions? Consider adding reference to the Appendix near the description of Table 2.

Line 356: “we observed that subjects tried to use an ankle strategy but quickly changed to hip strategy once the perturbation started, Figure 6”. In Figure 6, inversely, the subject is initially in the Hip strategy area and then at the perturbation onset goes to the Ankle strategy area. Which is correct?

Minor points:

Line 142 (description of the Pretest): the abbreviation BW is not defined.

Line 196 (section 2.1): the abbreviation COP is not defined.

In the Figure 1 caption, there is an unncecessary “up”: “Body diagram in our experimental setup up displaying belts”.

Line 163 – unnecessary “The”: “The In summary”

Figure 2: It would be much better to add a timescale or description of intervention procedure: how long (1 hour / one day / one week etc.) and how many repetitions?

Comments on the Quality of English Language

All good

Author Response

Thank you for your favorable review of our paper. We attempted to address the issues brought up by your major points and to make the corrections implied in your minor points. Please see the file attached.
